# OpenReview forum: "Beyond Worst-Case: Efficient Robust RL via In-Context Generalization"
_ICLR.cc/2026/Conference — Submitted to ICLR 2026_

### Official Review · Reviewer_g2Pv · 2025-10-29

**Soundness:** 1
**Presentation:** 1
**Contribution:** 2
**Rating:** 2
**Confidence:** 2

**Summary:**

In this work, the authors try to link robustness and generalization in in-context RL (ICRL). The authors propose a training procedure that augments the pretraining dataset with action labels obtained from robust policies, trained on diverse environment disturbances, so the transformer learns to generalize across tasks and perturbations.  The authors being by explaining the general idea of solving first a minmax game to obtain robust policies, and then use these robust policies to obtain action labels that are used as targets for training a DPT-like model. The authors then propose a series of relaxation to improve computational efficiency. They show   numerical results across different environments,  Dark Room, Meta-World, and MuJoCo tasks.

**Strengths:**

- I appreciate the efforts of the author to study the link between in-context learning and robustness
- The idea of reframing robust training as data/task augmentation over adversaries, avoiding possible computational burdens, can improve computational efficiency.
- Numerical evidence is provided on a range of environments, comparing to  DT-family and robust-RL baselines.

**Weaknesses:**

At the moment, there are 2 main weaknesses for me in the manuscript. First, the readability is quite low. I found it particularly slow and difficult to read the paper. There is redundant content, and the main ideas of the paper should be explained more clearly. Secondly, I am not convinced by the modeling, numerical results and the lack of theoretical results showing the robustness properties of the policy learned by the authors.

Regarding the presentation, the authors use a narrative style that probably hurts more than it helps. For example, the method part starts by explaining the general idea of solving first a minmax game to obtain robust policies, and then use these robust policies to obtain action labels to be used as targets train a DPT-like model. Then they try to alleviate the computational burden of the method by proposing a series of changes, and describe these changes in the subsequent 2 sections. However, the presentation is quite verbose, and mostly provided in textual form, making it hard to read, and by the time I reached the final part, I was confused about what the algorithm is really doing.

Here's a list of weaknesses


- I believe sentences of this type“By contrast, in-context RL (ICRL) has recently attracted considerable attention for its ability to generalize to novel tasks. In ICRL, we pretrain transformer models (TMs) on a diverse suite of RL tasks and then deploy them to novel ones.” are  a bit misleading. To my understanding, ICRL is a form of meta-learning, based on training a model on a prior set of tasks. Prior literature, to the best of my knowledge, does not necessarily show the capability of generalizing to task that do not belong to this prior set.
-	The introduction is quite verbose, and a reader gets lost easily. After reading it I was left more confused. The authors  try to motivate/describe what they do, but it cannot be precise because they haven’t introduced the proper notation/methodology.
-	Line 135: in in-context learning, where the task is drawn from a prior over tasks, the expected cumulative reward is averaged over the prior. The agent usually has no knowledge of the task faced. In eq. 1  the agent is conditioned on data from the data generated from $tau$, but not on $tau$ itself.
-	Some assumptions are missing. What assumptions do we have on the set of tasks $\mathcal{M}$ and on the prior $p_\tau$.
-	Line 180: it is a bit confusing to denote both policies by $\pi$, although the adversary’s policy is parametrized by $\phi$ (but the agent policy is not?)
-	I’m not convinced that solving eq. 3 solves for a “robust policy” as the authors state. It is indeed minimax with respect to opponents playing this Markov game, but then the kind of policy we get depends on the range of things the adversary is allowed to do. Hence, what capabilities does the adversary have? Does this framework capture problems where task-parameter are allowed to vary? (e.g., the gravity parameter in robotic tasks, friction coefficients, etc.). How is this related to robustness in the classical sense that “performance should not vary across a range of parameters”? Why should we assume that the adversary has the same information as the agent, and not more? (Or less?)
-	Section 3.2: here the authors introduce the concept of training robust policies for each task in the set of tasks so that they can then train another policy, in a supervised manner, by using as targets the robust actions predicted the policy trained in the first step. However, in this two-steps scenario I would expect to split the dataset of tasks between the first and the second phases to improve generalization. Moreover, it is also not clear why by doing so we obtain a policy that is robust. What is the theoretical grounding? (as a comparison, imitation learning in RL is not necessarily the best thing to do).
-	Section 4.1: the same concerns mentioned in the previous point also apply here. The authors however do not solve a min-max problem anymore, but fix a prior over adversaries. This seems effectively like domain randomization, but a policy here is trained on a (environment, adversary)-pair, so it’s not even averaged over the prior of adversaries, making it not scalable (if we have M environments and K adversaries per environment, we need to train MK policies). Why not apply directly a domain randomization approach?
-	Section 4.2: At the current state, the section is hardly readable, and I did not understand what the authors are trying to do. This was possibly the least clear part of the manuscript for me.
-	Formal theoretical results are missing. Informal ones are provided in the form of text. Moreover, the author state:
  - “Our first results reveal that well-pretrained ICAG models perform implicit Posterior Sampling “ This is known from prior literature that in-context learning is performing a sort of posterior sampling, the result is not  novel.
  - “By implicitly performing PS during deployment over both the deployment environment and potential adversarial perturbations, transformer models pretrained by ICAG are able to act optimally in the presence of an adversary capable of modifying the environment.” This is not clear that ICAG acts optimally. It’s also not clear why it should be robust. How is this proved? And does an adversary that modifies an environment fit the Markov game model proposed by the authors?
-	Regarding the numerical results there are some issues:
  - A comparison with CVaR-PPO is missing. Furthemore, why is M2TD3 not used also in Mujoco tasks?
  - The plots at the moment are barely readable
  - It’s not clear how robustness is showed from the results. For example, the authors could show how the performance varies for different parameters. Ideally, this performance should remain stable. clearly show how the performance varies

**Questions:**

- "deployment parity” is not well defined
-	Line 149, the terms offline/online are a bit confusing, especially if we compare to offline/online RL
-	Line 250: what is meant by parameter set here?
-	What is concretely the space $\Phi$

---

> ### Author Response · Authors · 2025-11-26
>
> Dear Reviewer g2Pv,
>
> We appreciate your thoughtful comments. Please see our response below, which we hope can address your concerns.
>
> ---
>
> > ```Prior literature, to the best of my knowledge, does not necessarily show the capability of generalizing to task that do not belong to this prior set. What assumptions do we have on the set of tasks and on the prior.```
>
> We respectfully clarify that, in the canonical formulation of ICRL [1,2], transformer policies are explicitly pretrained and evaluated on different sets of tasks. That is, **the test tasks are not observed during pretraining**.
>
> We agree that, to enable meta-learning or efficient generalization to these unseen tasks, prior work assumes that both training and testing tasks come from the **same task space** and are drawn from the **same underlying task distribution**. This assumption provides **shared structure across tasks**, thereby facilitating efficient generalization to unseen tasks. For instance, in the widely used DarkRoom benchmark, the agent is trained and tested on rooms with different goal positions, which constitutes a change of task despite belonging to the same task family.
>
> However, **in practice**, prior ICRL studies [1,2] consistently evaluate models on unseen tasks, and we follow this established setting. The goal here is to create a **challenging** evaluation protocol that thoroughly tests an ICRL model’s ability to generalize **beyond** the tasks encountered during pretraining (this setting is much more challenging than testing on the same tasks used for pretraining, thus more thoroughly testing the generalization capability of ICRL models).
>
> In summary, we use **consistent** problem setting, assumptions and experiment setup with prior works.
>
> > Manuscript Structure and Readability: ```I found it particularly slow and difficult to read the paper.```
>
> We understand your concern. Following your advice, we have updated the manuscript and trimmed our narratives.
> In particular, we add a summary paragraph in the introduction to highlight the structure of the manuscript. In addition, we also include a brief discussion at the beginning of the algorithm section (Section 4) to clarify the connections between different components.
>
> We would like to humbly **clarify the logic and flow of our work**, as all the components are closely related.
> - **Robust RL and the Conservative Policy Problem**: It is well-known that robust RL faces the challenge that the worst-case approach can lead to conservative policies with low nominal return. In other words, to ensure robustness, we need to sacrifice nominal performance and even performances under mild perturbations, as we are optimizing for the worst-case performance. This is not efficient in terms of performance.
> - **Contribution 1.** To tackle this challenge, our insight is to build upon the generalization capability of ICRL models to **transform robustness problems to generalization problems**. In particular, if the models (policies) can autonomously adapt to the testing environments, we don’t suffer from the conservative policy problem.
> - **Contribution 2.** We achieve this efficient adaptation by using ICRL models (DPT), which adapt to different environments **without** any parameter updates. Using the same training facilities (environment simulators with perturbing adversaries) as robust RL methods, we observe that ICRL models already **outperform** state-of-the-art robust RL methods, despite being pretrained **without** any robust RL objectives. We believe this is a valuable finding.
> - **Contribution 3.** We further realize that the already competitive performance of ICRL models can be further enhanced by increasing the robustness of ICRL models. To this end, we propose the ICAG algorithm. The interesting insight here is that, **unlike conventional robust RL, by transforming the robustness problem into the generalization problem, we can enhance the robustness of ICRL models without any worst-case (max-min) optimizations**, which are inefficient and difficult to solve.
> - **Contribution 4.** Noticing that the execution of ICAG can be time-consuming and expensive, we bootstrap on the existing generalization capability of ICRL models to efficiently collect new pretraining data, thereby increasing the scalability of ICAG. We term this efficient approach ICAA.
>
> As shown above, our contributions are closely connected and build on one another in sequence, which results in our narrative. The key **algorithmic deliverables** of this work are threefold:
> - **ICAG for robust RL**: ICAG can be applied as a robust RL algorithm that improves both nominal performance and performance under both mild and severe disturbances.
> - **ICAG for robust ICRL**: ICAG is also an efficient robust ICRL approach that substantially enhances ICRL models’ resilience and performance under disturbances.
> - **ICAA**: a lightweight and efficient implementation of ICAG.

---

> ### Author Response · Authors · 2025-11-26
>
> > ```Line 135: … The agent usually has no knowledge of the task faced … In eq. 1 the agent is conditioned on data from the data generated from \tau, but not on \tau itself.```
>
> We agree that the agent has no knowledge of the task faced, using only the data collected from $\tau$. In particular, line 135 is about MDP rather than in-context learning, and we use conditioning on $\tau$ to denote that the reward function and transition dynamics follow those of $\tau$. This is a standard notation, as also used in the referred work CVaR-PPO [13].
>
> In eq.1, we indeed used the notation $D^i$ to denote the data collected from task $\tau^i$ . The agent has **no** knowledge about $\tau^i$ itself.
>
> > ```Line 180: it is a bit confusing to denote both policies by $\pi$ , although the adversary’s policy is parametrized by $\phi$ (but the agent policy is not?)```
>
> Thank you for this feedback. To avoid confusion, **we have updated our manuscript to change the adversary’s notation** from $\pi_\phi^a$ to $\phi$, as our setup is not limited to parametric adversaries.
>
> > ```I’m not convinced that solving eq. 3 solves for a “robust policy” as the authors state. It is indeed minimax with respect to opponents playing this Markov game, but then the kind of policy we get depends on the range of things the adversary is allowed to do. Hence, what capabilities does the adversary have? Does this framework capture problems where task-parameter are allowed to vary? (e.g., the gravity parameter in robotic tasks, friction coefficients, etc.). ```
>
> We respectively clarify that the robustness objective in eq.3 was not stated by the authors ourselves. Instead, **this is an established training paradigm in robust RL** [2, 3, 5, 6, 7]. More generally speaking, while robustness can be an abstract concept,  in this work, we follow prior works to consider robustness as a policy property that **the policy maintains strong performance under disturbances or adversary attacks**. In this setting, max-min optimization is the **canonical** paradigm to achieve it [8-12].
>
> In settings like RARL [2], solving the max-min optimization problem in eq. 3 indeed leads to robust policies, as it guarantees maximum return against the worst-case disturbance within the defined uncertainty set or adversary parameter set. Notably, **this approach has been extensively proven to generate robust policies for problems where task-parameters (e.g., the gravity parameter in robotic tasks, friction coefficients, etc.) are allowed to vary [2, 3, 5, 7]**.
>
> Our evaluation also explicitly assesses robustness against environmental parameter variations (e.g., mass, friction) via the heatmaps in Figures 12~16, where ICAG also demonstrates strong performance.
>
> > ```Why should we assume that the adversary has the same information as the agent, and not more? (Or less?)```
>
> We follow the standard and most tractable assumption in adversarial RL that the adversary operates with the same observational information as the agent (the state s). We note this setting is **easily extendable** to scenarios where the adversary may have **different or more limited information**.
>
> > ```However, in this two-steps scenario I would expect to split the dataset of tasks between the first and the second phases to improve generalization```
>
> We appreciate the reviewer's query regarding task splitting. However, we respectfully point out that task splitting is **infeasible** here.
>
> Consider dividing the pretraining tasks into two disjoint sets, A and B. In principle, one might hope to (i) train robust policies on tasks in A, and then (ii) use these robust policies to generate action labels for tasks in B during ICRL pretraining. Unfortunately, this is not possible. Robust policies are **task-specific** (we train one robust policy per task) and a robust policy trained on a task in A cannot produce valid (let alone robust) action labels for a different task in B.
>
> As a result, pretraining on tasks in B still requires robust policies for tasks in B, which defeats the purpose of splitting. In other words, to label tasks in B, we must already have trained robust policies on B, making the proposed task split inherently infeasible.
>
> > ```Moreover, it is also not clear why by doing so we obtain a policy that is robust. What is the theoretical grounding?```
>
> This is a great question. To answer it, we rely on established theory demonstrating the capabilities of ICRL via supervised pretraining. Specifically, our pretrained robust policies (used for generating the pretraining dataset) can be referred to as the **context algorithm** in related literature [4], providing theoretical grounding for this approach.
>
> On a high level, when pretrained with the objectives in Section 3.2, the ICRL models are essentially pretrained to estimate the policy with the best worst-case performance for a given adversary/uncertainty set whereas in the canonical setting of ICRL, they estimate the policies with the optimal nominal performance.

---

> ### Author Response · Authors · 2025-11-26
>
> > ``` This seems effectively like domain randomization, but a policy here is trained on a (environment, adversary)-pair, so it’s not even averaged over the prior of adversaries, making it not scalable (if we have M environments and K adversaries per environment, we need to train MK policies). Why not apply directly a domain randomization approach?```
>
> While ICAG (In-context Adversarial Generalization) shares commonality with Domain Randomization (DR) by leveraging varied environments, it is fundamentally distinct and superior. DR aims for a **single, static policy** robust to a distribution of dynamics, whereas ICAG trains a **meta-policy capable of in-context adaptation to different tasks $\tau$ and disturbances $\phi$**. The core difference lies in the mechanism: ICAG is proven (Theorem F.3) to perform implicit Posterior Sampling (PS) over the joint space of $\tau$ and $\phi$ during deployment, which is **a dynamic and sample-efficient adaptation superior to the static policy learned by DR**. This leads to **significantly enhanced performance** over state-of-the-art static policy algorithms including DR and M2TD3, as shown in our extensive experiments.
>
> In terms of **scalability**, we train one single transformer model​ on the aggregated dataset from M×K distinct variation environments. The relatively high computational burden is the **one-time** cost of generating the pretraining dataset. With that being said, the data generating process over MxK variations environments is **readily parallelizable** for efficiency (as the SAC runs are independent across all the MxK variation environments) while the conventional max-min optimization needs to be done sequentially and cannot be parallelized.
>
> Lastly, we highlight that one can always choose **ICAA as an economic alternative**, requiring much less computation and rollouts yet still demonstrating strong performance.
>
> > ```Section 4.2: … I did not understand what the authors are trying to do.```
>
> We understand this concern. To address this, we included an extra paragraph at the beginning of Section 4 to help connect our contributions. In addition, we include our algorithms’ pseudocode in Section 4.2 to help understand them.
>
> The structural purpose of ICAA is to serve as a **computationally efficient and scalable alternative to ICAG**, addressing the data acquisition cost of ICAG. The core idea is **recursive, self-generated label bootstrapping**: instead of training an optimal RL policy from scratch, ICAA leverages a moderately pretrained transformer ($T_{\theta}^j$​) to quickly adapt to new adversarial environments online, then uses the resulting high-quality trajectories as improved action labels for continued supervised pretraining  to have an improved model ($T_{\theta}^{j+1}$​). This iterative process enhances robustness efficiently, as **empirically validated** by our ablation study in Figure 8.
>
> > ```Formal theoretical results are missing…This is known from prior literature that in-context learning is performing a sort of posterior sampling, the result is not novel```
>
> **Due to space constraints, we moved the formal results to Appendix F and only left informal ones to motivate better understanding of our algorithms**.
>
> While it is known that ICRL in the canonical setting is conducting a sort of posterior sampling, **we extend these theoretical results to our robustness setting and proposed algorithms**. These results are valuable in terms of rigorously understanding why the proposed algorithm ICAG and ICAA can indeed generate robust actions and outperforms conventional RL methods.
>
> > ```This is not clear that ICAG acts optimally. It’s also not clear why it should be robust. How is this proved? And does an adversary that modifies an environment fit the Markov game model proposed by the authors?```
>
> Yes. An adversary that modifies an environment is a **concrete instantiation** of the Markov game model. In general, if we choose the adversary capable enough, we can simulate any type of disturbances of interest including varying task-parameters. This is exactly why adversarying training can generate robust policies.
>
> Our theoretical results in F.3 proves that, when deployed to an environment with disturbances, ICAG leads to pretrained ICRL models that conduct posterior sampling (PS) jointly over the environment and the adversary. As PS is known to be most efficient algorithm for online adaptation, **ICAG increases its performance in the deployed environment with optimal sample complexities**, implying that ICAG acts optimally.
>
> By **robustness**, we mean that **the policy maintains strong performance under disturbances**, and our theoretical verifies that ICAG has this property, as ICAG essentially uses PS to adapt to any given deployment environment and disturbances, thereby **outperforming static policies.** This is also **verified by our extensive experiments where ICAG consistently demonstrates strong performances with and without disturbances.**

---

> ### Author Response · Authors · 2025-11-26
>
> > ```A comparison with CVaR-PPO is missing. Furthemore, why is M2TD3 not used also in Mujoco tasks?```
>
> **We follow your advice to  integrate M2TD3 into all MuJoCo tasks**, which now serves as a strong, robust baseline across our entire benchmark suite in Figure 3 and Figure 12~17. **Our methods significantly outperform M2TD3 on MuJoCo tasks**, an observation **consistent** with results on Dark Room and Meta World.
>
> **We also added CVaR-PPO to the locomotion tasks (Quadruped and Cheetah)** in Figure 1(b), **further validating our performance against risk-aware robust methods**. This expansion ensures our MuJoCo results now include a comprehensive set of robust baselines, confirming the superior performance and robustness of our proposed method.
>
> > ```What is meant by parameter set here? What is concretely the space \Phi```
>
> The space $\Phi$ represents the pre-defined adversary space or uncertainty set for a task $\tau$. Concretely, this is the set of constraints, limits, and disturbance policies that the adversary is permitted to use against the agent. When assuming a parametric adversary, we can call this space the parameter set of adversary. Our definition and setup of the adversary parameter set directly follow the classical robust RL approach using adversarial training [2, 3]. The exact composition of $\Phi$ is task-dependent; for instance, in MuJoCo, the adversary set is parameterized to build the adversarial action space and maximum force bounds as detailed in Table 2 in appendix.
>
> > ```Line 149, the terms offline/online are a bit confusing, especially if we compare to offline/online RL```
>
> We thank the reviewer for the feedback on clarity, but we humbly argue that our usage of Offline/Online Deployment is **consistent** with established In-Context RL (ICRL) literature [1]. **This usage is explicitly defined right after we mentioned offline/online terms and visually supported by the procedure in Algorithm 3 in appendix**. In particular, the offline/online deployments align with offline/online RL, as both offline deployment and RL use trajectories collected by behavioral policies while  both online deployment and RL rely on experiences gathered by agents themselves. We believe aligning our terminology with the conventions of the ICRL community resolves the ambiguity.
>
>
> > ```The plots at the moment are barely readable ```
>
> Thank you for this advice.
>
> To improve visual clarity, we have reorganized all composite figures in the original main manuscript (Figures 3, 4, and 5) from four sub-plots in one row to a more spacious and scannable arrangement of three sub-plots per row.
>
> > ```It’s not clear how robustness is showed from the results. For example, the authors could show how the performance varies for different parameters. Ideally, this performance should remain stable. clearly show how the performance varies```
>
> We confirm **this specific analysis was already included in our original manuscript** in the form of heatmaps. To aid verification, we direct the reviewer to Figure 12~17.
>
> >  ```"deployment parity” is not well defined```
>
> We provide an **explicit definition** of *Deployment Parity* right after we propose this term. Specifically,  Deployment Parity refers to the critical operational similarity that both classical robust RL policies and pretrained ICRL models require no further parameter updates at test time (deployment). **We have highlighted this in the manuscript to avoid confusion.**

---

> ### Author Response · Authors · 2025-11-26
> **References**
>
> [1] Lee et al., “Supervised Pretraining Can Learn In-Context Reinforcement Learning”, NeurIPS 2023
>
> [2] Pinto et al., “Robust Adversarial Reinforcement Learning”, ICML 2017
>
> [3] Dong et al., “Variational Adversarial Training Towards Policies with Improved Robustness”, AISTATS 2025
>
> [4] Lin et al., “Transformers as Decision Makers: Provable In-Context Reinforcement Learning via Supervised Pretraining”, ICLR 2024
>
> [5] Reddi et al., “Robust Adversarial Reinforcement Learning via Bounded Rationality Curricula”, ICLR 2024
>
> [6] Huang et al., “Robust Reinforcement Learning as a Stackelberg Game via Adaptively-Regularized Adversarial Training”, IJCAI 2022
>
> [7] Vinitsky et al., “Robust Reinforcement Learning using Adversarial Populations”, 2020
>
> [8] Gorissen et al., “A practical guide to robust optimization”, Omega 2015
>
> [9] Bertsimas et al., “Theory and applications of Robust Optimization”, 2010
>
> [10] Shafiee et al., “Developing Bidding and Offering Curves of a Price-Maker Energy Storage Facility Based on Robust Optimization”, IEEE Transactions on Smart Grid, 2019
>
> [11] Qiu et al., “Multistage Mixed-Integer Robust Optimization for Power Grid Scheduling: An Efficient Reformulation Algorithm”, IEEE Transactions on Sustainable Energy, 2023
>
> [12] Hu et al., “Robust SCUC Considering Continuous/Discrete Uncertainties and Quick-Start Units: A Two-Stage Robust Optimization With Mixed-Integer Recourse," in IEEE Transactions on Power Systems, 2023
>
> [13] Ying et al., “Towards Safe Reinforcement Learning via Constraining Conditional Value-at-Risk”, IJCAI 2022

---

### Official Review · Reviewer_cbu6 · 2025-10-31

**Soundness:** 3
**Presentation:** 2
**Contribution:** 3
**Rating:** 6
**Confidence:** 4

**Summary:**

This paper studies robustness in in-context RL and proposes ICAG/ICAA as scalable alternatives to explicit max–min robust training.
The idea is to turn robustness into a generalization problem: instead of labeling with robust actions from a per-task max–min solve (IC2RL), the authors augment each task with K adversarial/perturbed variants and label with the optimal action in each variant, then pretrain a transformer that learns to generalize across these disturbances.
Concretely, the adversary set is constructed per benchmark: in Dark Room, robustness is probed by action overrides at probability p and by switching among several OOD priors; in Meta-World, robustness is tested under state disturbances; in MuJoCo, the authors instantiate K fixed adversary policies, each a distinct neural network, and train SAC to convergence under each adversary to collect trajectories; yielding variation environments $(\tau_i,\phi_{i,k})$ used for supervised pretraining; robustness is then evaluated both against adversarial disturbances and under environmental parameter shifts (mass/friction).
Across navigation, manipulation, and continuous control, ICAG/ICAA report higher nominal and disturbed performance than strong ICRL and robust-RL baselines, while avoiding explicit worst-case optimization.

**Strengths:**

* Conceptual reframing: Treating robustness as in-context generalization is elegant and practical; ICAG avoids the computational and pessimism issues of explicit max–min solves.
* Experiments: Results span Dark Room, Meta-World, and MuJoCo; ICAG/ICAA consistently outperform DPT/DT variants and are competitive with or better than robust-RL baselines (RARL, QARL) in disturbed settings.
* Positioning vs. robust DT: The paper situates itself against ARDT, a worst-case-aware DT that learns minimax returns via expectile regression; this contextualization clarifies the difference between learning worst-case labels (ARDT) and generalizing across adversaries (ICAG).

* No max–min in label generation + parallelizability. ICAG labels come from per-variant optimal policies $\pi^*_{\tau,\phi}$ (Eq. 5), not a minimax solve (except for MuJoCo); training these $K$ variants is embarrassingly parallel, which is a pragmatic advantage over robust max–min pipelines.

* "Null adversary" design prevents over-pessimism. By including $\phi_0$ (no disturbance) in $\Phi(\tau)$, the augmented space $M_v$ strictly contains the original task space $M$, so robustness training doubles as principled data augmentation without sacrificing nominal performance, which is the main problem in robust reinforcement learning.

* The method connects ICRL pretraining to posterior sampling  and gives a monotonic improvement result for ICAA, grounding the “generalize-via-variation” design in a principled lens familiar from DPT/ICRL theory.

* The construction naturally covers agent disturbance (fixed adversaries for worst-case testing) and environmental change (mass/friction grids), aligning the training data recipe with the evaluation metrics

**Weaknesses:**

* IC2RL label quality not evaluated: While it’s clear in principle that IC2RL’s performance is bounded by the quality of the robust policy used to label, there is no ablation varying robust-policy quality (e.g., early-stopped RARL vs well-converged, or RARL vs stronger solvers) to measure downstream IC2RL sensitivity.
* MuJoCo baselines miss an important robust method: The MuJoCo section compares against SAC/RARL/QARL and DT baselines, but does not include M2TD3  even though M2TD3 is used elsewhere in the paper and in recent robust-RL benchmarks; this weakens claims on continuous-control robustness.
* Main text not self-contained: Important setup and full results are pushed to appendices (C–E), making it harder to assess choices like adversary architectures, K, compute budget, and per-task outcomes directly from the main body.
* Compute cost for MuJoCo data: Even without explicit max–min, training SAC to convergence for each of K adversaries per task is expensive (and scales linearly with K×tasks), which may limit practicality in larger suites; the paper would benefit from clearer accounting of this cost.

**Questions:**

1. IC2RL sensitivity to labeler quality: Can you report an ablation that varies the robust labeler’s quality (e.g., early vs late RARL/QARL checkpoints, or alternative robust solvers) and measures the downstream IC2RL performance? This would directly test the hypothesized dependence.
2. Why no M2TD3 on MuJoCo? You include M2TD3 in Dark Room/Meta-World figures but omit it for MuJoCo locomotion/adversarial results, where it is a strong and less over-conservative robust baseline. Could you add M2TD3 (e.g., via RRLS-style uncertainty sets) to your MuJoCo comparisons?
3. Adversary-set design details: How should practitioners choose K and the adversary architectures (capacity, action spaces, force bounds) across tasks? Do results saturate with K (e.g., 4/8/16), and what is the failure mode if adversaries are too weak/strong? Some guidance beyond the current appendix pointers would help.
4. Could you provide wall-clock/time-per-task and total pretraining cost for SAC×K adversary training and dataset generation, alongside baselines (RARL/QARL, ARDT)? This would clarify “efficiency” claims.
5. Worst-case metric definition. For MuJoCo, are “with disturbance” scores a min over adversary policies or an average? A per-task table reporting mean and worst-case (min) across fixed adversaries would make the robustness metric explicit.

---

> ### Author Response · Authors · 2025-11-26
>
> Dear Reviewer cbu6,
>
> We appreciate your constructive comment. Please see our response to your concerns below.
>
> ---
>
> > ```IC2RL label quality not evaluated: While it’s clear in principle that IC2RL’s performance is bounded by the quality of the robust policy used to label, there is no ablation varying robust-policy quality to measure downstream IC2RL sensitivity.```
>
> Thank you for this suggestion. Following your advice, we include **an additional ablation study in Figure 9 and additional discussion in  Appendix E.3**.
> To measure the sensitivity of the downstream IC2RL model, we used RARL policies trained to half of the total learning steps until convergence (early-stopped) as a low-quality label source, and compared it against the original IC2RL using the well-converged RARL policies. **As expected, the IC2RL model trained with the early-stopped labels showed significantly degraded performance**, confirming that the performance of IC2RL is fundamentally bounded by the quality and high computational cost of its robust label-generating policy. This result validates our core motivation in pursuing ICAG and ICAA as more scalable alternatives that avoid this cost and reliance.
>
> > ``` The MuJoCo section compares against SAC/RARL/QARL and DT baselines, but does not include M2TD3```
>
> **We follow your advice to  integrate M2TD3 into all MuJoCo tasks**, which now serves as a strong, robust baseline across our entire benchmark suite in Figure 3 and Figure 12~17. **We observe our methods notably outperforms M2TD3 as well**.  In addition,  **we also added CVaR-PPO to the locomotion tasks (Quadruped and Cheetah)** in Figure 1(b), further validating our performance against risk-aware robust methods. This expansion ensures our MuJoCo results now include a comprehensive set of robust baselines, confirming the superior performance and robustness of our proposed method.
>
> > ```Main text not self-contained```
>
> We thank the reviewer for this constructive suggestion.
>
> We have gladly used the additional page for the rebuttal to improve the paper's self-containedness. **We moved the most critical information from Appendices C-E into the main body**. Specifically, we incorporate our key adversary design choices and number of adversaries $K$ in Sections 5.1~5.3. We also include an additional paragraph for Computation and Implementation Efficiency. We are confident these additions will provide a clearer view of our experimental setup directly within the main text.
>
> > ```Even without explicit max–min, training SAC to convergence for each of K adversaries per task is expensive…the paper would benefit from clearer accounting of this cost. Could you provide wall-clock/time-per-task and total pretraining cost for SAC×K adversary training and dataset generation, alongside baselines (RARL/QARL, ARDT)? ```
>
>
> We understand this concern and thank the reviewer for highlighting the practical cost of our ICAG data generation. **This is the exact reason why we proposed ICAA,** which bootstraps on the existing generalization capability of ICRL models to efficiently increase their robustness with minimal trajectory rollouts. As shown in our ablation study (in Figure 8), **ICAA quickly improves model performance, significantly reducing the total number of rollouts and computation cost**.
> More importantly, we kindly note that the data generation process for ICAG is **readily parallelizable**, making them suitable for modern high-computing GPUs.
>
> **Wall-Clock Time Accounting.** Following your advice, we account for the cost of SAC data generation. The table below provides the average wall-clock time required for training policies per task for quadruped environment in MuJoCo (for dataset generation and baseline comparison). The ICAG data collection process utilizes SAC runs, which are **significantly faster than the maxmin baselines** (RARL and QARL) and ARDT's complex objective.
>
> | Method                          | Optimization Type            | Average Wall-Clock Time Per Task (Min) |
> |---------------------------------|------------------------------|-----------------------------------------|
> | **SAC** (ICAG Dataset Basis)     | Maximize                     | 29                                      |
> | **QARL** ($\max \min$ Variant)   | Quantal Response Equilibrium | 46                                      |
> | **RARL** ($\max \min$ Game)      | Minimax Game                 | 52                                      |
> | **ARDT** (Robust DT Baseline)    | Minmax Expectile Regression  | 115                                     |
>
>
> These runtimes confirm that ICAG is at least two times faster than IC2RL, significantly reducing the total runtime with considerably increased performance. This confirms that ICAG achieves both runtime and performance efficiency.
>
> Lastly, we note that ICAA serves as a complementary method with much less computation when computation budget is low and parallelism is infeasible.

---

> > ### Author Response · Authors · 2025-11-26
> >
> > > ```Adversary-set design details: How should practitioners choose K and the adversary architectures (capacity, action spaces, force bounds) across tasks? Do results saturate with K (e.g., 4/8/16), and what is the failure mode if adversaries are too weak/strong? ```
> >
> > We appreciate this constructive question.
> >
> > **Choice of K.** In *Appendix E.1 (Figure 7)*, our results on Dark Room show that performance on held-out tasks **does saturate** with increasing K demonstrating diminishing returns. We observe that K=8 provides a good balance of performance and efficiency. We also show a similar trend in MuJoCo (Figure 9) where ICAG10 outperforms ICAG1 and ICAG5. As DarkRoom is a challenging benchmark and K should be **positively correlated** with the task complexity, we believe **$K=8$ is a good choice for most real-world tasks, and practitioners can choose a larger K for more complex tasks** (e.g., tasks for which SAC agents take more steps to converge).
> >
> > **Insights and Suggestions for Adversary Set.** We kindly note the following general guidance and failure modes:
> > - If adversaries are too weak: The robustness benefit will be minimal, and the model's performance will be close to the standard, non-robust baseline (like DPT).
> > - If adversaries are too strong: They can make the pre-training tasks unsolvable. This would fill the pre-training dataset with low-quality, zero-reward trajectories, providing a poor learning signal and harming both nominal and robust performance.
> >
> > Therefore, practitioners should aim for a diverse set of adversaries that are challenging but not impossible for a base policy to learn against. **For adversary architectures and force bounds, we recommend following established practices from robust RL literature for the specific domain**. For instance, in our MuJoCo experiments, we calibrated adversary forces to be significant enough to foster robustness while still being solvable, as detailed in Appendix D.3 .
> >
> > > ```For MuJoCo, are “with disturbance” scores a min over adversary policies or an average? A per-task table reporting mean and worst-case (min) across fixed adversaries would make the robustness metric explicit.```
> >
> > We would like to clarify that  the "Return under Adversary" scores in our MuJoCo plots (Figs 10-12) are neither a "min" nor an "average" over the K=10 fixed adversaries used during pre-training.
> >
> > Instead, we used a more rigorous evaluation to find a true "worst-case" performance for each method. For each final trained agent (the "protagonist"), our evaluation process includes:
> > - We froze the protagonist's policy after its training was complete.
> > - We then trained a new adversary policy from scratch to play against this frozen protagonist until the adversary converged (i.e., it learned the best-possible counter-strategy).
> > - This newly converged, best-response adversary is what we define as the "worst-case adversary" for that specific agent.
> >
> > Thus, the scores in the plots represent the **policies’ performance under the worst-case disturbance**, i.e., protagonist's performance against its own, specialized "worst-case adversary." The box plots show the distribution of these scores, as this entire procedure (training the protagonist, then training its worst-case adversary) was repeated for 10 random seeds. **We will revise Appendix E.4 to make this evaluation process explicit**, as it's a **stronger metric** than what was assumed and was commonly used in robust RL methods [1, 2, 3].

---

> ### Author Response · Authors · 2025-11-26
> **References**
>
> [1] Pinto et al., “Robust Adversarial Reinforcement Learning”, ICML 2017
>
> [2] Reddi et al., “Robust Adversarial Reinforcement Learning via Bounded Rationality Curricula”, ICLR 2024
>
> [3] Dong et al., “Variational Adversarial Training Towards Policies with Improved Robustness”, AISTATS 2025

---

### Official Review · Reviewer_Fyqe · 2025-10-31

**Soundness:** 3
**Presentation:** 3
**Contribution:** 3
**Rating:** 6
**Confidence:** 3

**Summary:**

This paper proposes a novel perspective on robust reinforcement learning by linking robustness to generalization in in-context reinforcement learning (ICRL). Rather than relying on computationally expensive worst-case minimax optimization, the authors demonstrate that robustness can emerge naturally from ICRL's generalization capabilities. The paper makes two key observations: (1) pretrained ICRL models (without explicit robustness training) already outperform dedicated robust RL baselines in classical robust RL settings, yet (2) these same models still exhibit significant performance degradation under deployment disturbances. To address this gap, the authors propose two complementary methods: In-Context Adversarial Generalization (ICAG), which augments pretraining with adversarial task variations to enable the transformer to generalize across disturbances without solving max-min problems, and In-Context Adversarial Adaptation (ICAA), which efficiently bootstraps high-quality action labels for continued pretraining using online deployment trajectories. Theoretical analysis shows that ICAG implicitly performs Posterior Sampling over both environments and adversaries (Theorem F.3), while ICAA provably improves action label quality across iterations (Theorem F.5). Empirical evaluation across sparse-reward navigation (Dark Room), manipulation (Meta-World), and continuous control (MuJoCo) demonstrates consistent improvements in both robustness and nominal performance compared to robust RL baselines and competing ICRL methods.

**Strengths:**

**Novel and well-motivated link between robustness and generalization.** The paper articulates a compelling insight that robustness can emerge from generalization rather than requiring explicit robust optimization. The observation in Figure 1 that ICRL methods substantially outperform dedicated robust RL methods (Domain Randomization, M2TD3, RARL, QARL) on classical robust RL benchmarks is striking and non-obvious, motivating the entire research direction.

**Principled theoretical analysis.** Theorems F.3 and F.5 provide formal guarantees connecting ICAG to Posterior Sampling and characterizing ICAA's iterative improvement. The connection to PS is particularly valuable, as PS is known to be sample-efficient and provides a theoretical justification for ICAG's effectiveness beyond heuristic intuition. The theoretical framework grounding the approach enhances credibility.

**Thoughtful treatment of deployment requirements and limitations.** Section 7 candidly acknowledges that ICRL requires context trajectories at deployment, which may be impractical in data-scarce safety-critical domains, and suggests when classical robust RL remains preferable.

**Weaknesses:**

**Limited novelty of core components.** While the link between robustness and generalization is novel, the individual techniques are not: augmenting environments with perturbations is standard data augmentation, and leveraging pretrained models to generate action labels for continued pretraining is established practice. The contribution is more one of combination and insight than technical innovation.

**Restrictive theoretical assumptions.** The theoretical analysis in Appendix G imposes strong conditions: (1) the joint distribution in Equation 6 assumes optimal action labels are available for each variation environment which is unrealistic in practice, (2) Assumptions F.1-F.4 in Appendix F require conditions like invertible Hessians and strict local maxima that may not hold for neural network policies. The gap between these assumptions and practical deep RL is significant, limiting theoretical guarantees' applicability.

**Disconnect between theory and practice.** ICAG pretraining uses the ideal supervised objective in Equation 1, but in practice, obtaining optimal action labels for every variation environment (τ, ϕ) via training policies to convergence is computationally expensive. While ICAA partially addresses this by using online-collected labels, the method introduces additional approximation that weakens theoretical guarantees. The practical pretraining cost is not thoroughly analyzed.

**Questions:**

**Q1:** What explains the remaining performance gap between ICAA and ICAG? Since ICAA uses non-optimal action labels, how much performance is sacrificed? Can tighter upper bounds be established?

**Q2:** How do ICAG and ICAA perform in sparse-action environments or with discrete action spaces? Experiments focus on continuous control; scalability to discrete domains is unclear.

---

> ### Author Response · Authors · 2025-11-26
>
> Dear Reviewer Fyqw,
>
> Thank you for your thoughtful comments. We hope our responses below can address your concerns.
>
> ---
>
> > **Contribution Clarification**: ```Restrictive theoretical assumptions…Disconnect between theory and practice. ```
>
> We appreciate the reviewer’s observation that some of our theoretical assumptions may not fully hold in complex real-world deep RL settings. Indeed, these assumptions are used primarily to provide **conceptual clarity** and a **rigorous foundation for understanding** why our approach works, rather than to mirror every detail of practical neural network training.
>
> That said, we emphasize that our **core** contribution is the insight, **not** the specific technical assumptions. Our insight reveals a fundamental linkage between **robustness** and **generalization**, from which two key takeaways emerge:
> - **ICRL models, despite being pretrained without any robustness objectives, already behave as strong robust RL algorithms** due to their inherent generalization structure.
> - **Unlike conventional robust RL, we can leverage this generalization capability to improve ICRL robustness without solving a difficult max–min optimization**, enabling a far more efficient and scalable solution.
>
> Guided by these insights, we introduce ICAG, a *simple yet effective* algorithm that improves both (i) the robustness of ICRL models and (ii) their performance across robustness benchmarks. Importantly, **the practical aspects of ICAG do not rely on the theoretical assumptions in a brittle way**; instead, they are **validated empirically**. In Figure 8, we also conduct an **extra ablation study** to validate that ICAA’s performance indeed quickly increases with more iterations in real-world environments where assumptions may fail to hold.
>
> We therefore complement the theoretical results (despite their assumptions) with **extensive experiments and ablation studies**, demonstrating that the practical algorithm remains effective even when theoretical assumptions are relaxed or only approximately met.
>
> > ```The contribution is more one of combination and insight than technical innovation.```
>
> We thank the reviewer for recognizing the novelty of our core insight: linking robustness and generalization. We agree that our approach's strength lies in this conceptual reframing and the effective combination of established components. However, rather than as drawbacks, **we humbly view the resulting simplicity of our methods as a key advantage**, making them practical and easy to apply, which is supported by our strong empirical results
>
> **Technical Novelty within ICRL Literature.** While the individual ideas (augmentation, iterative training) are known, it is still novel to apply them for **ICRL**. To the best of our knowledge, ICAA is the first to demonstrate how an iterative, self-labeling approach can be used to efficiently bootstrap robustness in the in-context RL setting, leveraging the model's own generalization capabilities to create high-quality labels for perturbed tasks. In addition, by treating ICRL robustness as a pure generalization problem over a space of environment-adversary pairs, we completely bypass the computationally expensive and pessimistic max-min optimization inherent in traditional robust RL.
>
> > ```What explains the remaining performance gap between ICAA and ICAG? Since ICAA uses non-optimal action labels, how much performance is sacrificed? Can tighter upper bounds be established?```
>
> This is a very insightful question. ICAA builds on the generalization ability of ICRL models to progressively collect action labels with higher qualities for pretraining. **If the ICRL models converges to the optimal policies, ICAA should match the performance of ICAG**. However, despite ICRL models’ strong performance, they are still suboptimal (*we use SAC policies trained  until convergence to collect action labels for ICAG, and these converged SAC policies are likely to outperform the ICRL models as they are trained using tens of thousands of trajectories for **each** environment while ICRL models only use tens of trajectories*), causing the gap between ICAA and ICAG. The regret bound here should be equivalent to the regret bound of ICRL models, which some prior work such as [1] has studied.
>
> [1] Lin et al., “Transformers as Decision Makers: Provable In-Context Reinforcement Learning via Supervised Pretraining”, ICLR 2024

---

> ### Author Response · Authors · 2025-11-26
>
> > ```The practical pretraining cost is not thoroughly analyzed.```
>
> **Economic Generation Cost of ICAA.** We kindly note that ICAA, in general, **does not** require a large amount of generation, as **its performance increases quickly in each iteration and usually saturates/reaches strong performance within 5 iterations**.
>
> **Ablation Study.** In Figure 8, we conduct an extra ablation study on ICAA for different iterations, demonstrating  its **fast performance improvement** over iterations and its quick saturation.
>
> **Pretraining Cost of ICAG.** The cost of collecting the ICAG pretraining dataset scales linearly with the total number of variation environments as $\mathcal{O}(MK)$. However, the total runtime/expense can mitigated by two factors:
> - *Complexity Reduction*: ICAG replaces the numerically complex and sequential maxmin optimization with a simpler, faster max optimization for each variation.
> - *Parallelization*: Crucially, the entire cost of the data generation process consists of **independent SAC runs** and is therefore **readily parallelizable**. This dramatically reduces the total wall-clock time, making the total pretraining cost amortized and feasible.
>
> **Wall-Clock Time Accounting.** To more thoroughly study the cost, the tables below provide the average wall-clock time required for training policies per task for a quadruped environment in MuJoCo (for dataset generation and baseline comparison). The ICAG data collection process utilizes SAC runs, which are significantly faster than the maxmin baselines (RARL and QARL) and ARDT's complex objective.
>
> | Method                          | Optimization Type            | Average Wall-Clock Time Per Task (Min) |
> |---------------------------------|------------------------------|-----------------------------------------|
> | **SAC** (ICAG Dataset Basis)     | Maximize                     | 29                                      |
> | **QARL** ($\max \min$ Variant)   | Quantal Response Equilibrium | 46                                      |
> | **RARL** ($\max \min$ Game)      | Minimax Game                 | 52                                      |
> | **ARDT** (Robust DT Baseline)    | Minmax Expectile Regression  | 115                                     |
>
>
> These runtimes confirm that ICAG is at least two times faster than IC2RL, significantly reducing the total runtime with considerably increased performance. This confirms that ICAG achieves both runtime and performance efficiency.
>
> Lastly, we note that ICAA serves as a complementary method to ICAG with much less computation when computation budget is low and parallelism is infesaible.
>
> > ```How do ICAG and ICAA perform in sparse-action environments or with discrete action spaces? Experiments focus on continuous control; scalability to discrete domains is unclear.```
>
> We appreciate the reviewer’s insightful suggestion.
>
> We note that **our original submission already includes experiments on environments with discrete action spaces**. Specifically, the **DarkRoom** task (described with details in Appendix D.1) is a discrete-action environment with five possible actions: move left, move right, move up, move down, and stay.
>
> As shown in the main results (Figure 3), **both ICAG and ICAA perform strongly in this discrete-action domain**, demonstrating that our methods are not limited to continuous-control settings. This provides empirical evidence that the proposed approach scales effectively to discrete-action environments.

---

### Official Review · Reviewer_C1rM · 2025-11-01

**Soundness:** 2
**Presentation:** 2
**Contribution:** 3
**Rating:** 6
**Confidence:** 3

**Summary:**

This paper exposes an interesting connection between robustness and generalization in in-context reinforcement learning (ICRL) that these two qualities are not orthogonal. In the first experiment, the authors show that ICRL without robustness objective can outperform classical robust RL baselines, indicating that ICRL is a preferable method since it avoids performance sacrifices---a typical challenge in robust RL.

**Strengths:**

The empirical observation of the connection between generalization and robustness is interesting. The experiments are conducted thoroughly. The proposed method is also simple, which is easy to apply.

**Weaknesses:**

1. **Overclaims**: In Fig 1(b), the authors draw the conclusion of ICRL better than classical RL. The authors should demonstrate these with more environments, not only just Dark Room. Otherwise, the claim that ICRL methods substantially outperform robust-RL baselines without robustness objective is overly strong. To me, addressing the robustness of ICRL is already a good motivation, even without claiming ICRL is better than classical robust RL in the first place.

2. **Presentation**: I think the authors should put the pseudo-code (short version) in the main text to help understand your algorithm. The texts from 316 to 330 are really hard to follow. The removable/reducible part to me is the theoretical analysis in 4.2 and some of the analysis in 4.1.

**Questions:**

Q: This question could be due to the unclear presentation, but from my understanding ICAA requires extensive generations from ICRL model. Will this computational cost be concerning?

---

> ### Author Response · Authors · 2025-11-26
>
> Dear Reviewer C1rM,
>
> Thank you for your helpful comments. We hope our responses below can address your concerns.
>
> ---
>
> > ```In Fig 1(b), the authors draw the conclusion of ICRL better than classical RL. The authors should demonstrate these with more environments, not only just Dark Room.```
>
> This is a great suggestion.
>
> In our original manuscript, while we focus on presenting the result for **Quadruped and Cheetah benchmarks**  in Figure 1 in the introduction, in the experiment section (Section 5) we also compare competitive ICRL models (DPT and ours) with strong robust RL baselines on **Dark Room** and **Meta World** (Figure 3). **Across all three benchmarks, we confirm the consistency of the observation that ICRL models are strong robust RL policies.**
>
> > ```To me, addressing the robustness of ICRL is already a good motivation, even without claiming ICRL is better than classical robust RL in the first place.```
>
> Our intention was to highlight the surprising observation that **standard ICRL models, which are designed for generalization, performed strongly on these classical robustness benchmarks even without an explicit robustness objective**, even consistently outperforming SOTA robust RL methods. This interesting link between generalization and robustness is a **core** part of our work's motivation. We believe this observation and insight are important and valuable to the community.
>
> However, we agree with the reviewer that our original phrasing may have been overly strong. We want our claims to be precise and conservative. Therefore, **we have revised our manuscript** to remove the word "substantially" when describing this result. We believe this makes the statement more mild and accurate, while still preserving the observation that motivated our investigation.
>
> > ```I think the authors should put the pseudo-code (short version) in the main text to help understand your algorithm.```
>
> Thank you for this great suggestion. Following your advice, **we have moved the pseudocode to the main text and trimmed the theoretical analysis in Section 4**, moving most of the details to the appendix.
>
> > ```This question could be due to the unclear presentation…Will this computational cost be concerning?```
>
> We appreciate and understand this concern. To improve clarity of our algorithm, we included the algorithm pseudocode into the main text and added additional explanation.
>
> **Economic Generation Cost of ICAA.** We kindly note that ICAA, in general, does not require a large amount of generation, as **its performance increases quickly in each iteration and usually saturates/reaches strong performance within 5 iterations**.
>
> **Ablation Study.** In Figure 8, we conduct an extra ablation study on ICAA for different iterations, demonstrating the **fast performance improvement over iterations and its quick saturation**.
>
> **Parallelization.** More importantly, we note that these rollouts can be readily parallelized, further reducing the running time.

---

### Author Response · Authors · 2025-12-03
**Rebuttal Summary**

Dear Reviewers and Area Chairs,

We thank all reviewers for their thoughtful evaluations. In particular, we deeply appreciate all the supports from the review committee:
- Reviewers C1rM, Fyqe, and cbu6 affirm that our paper investigates **an important and timely direction**, offers an **interesting and novel insight connecting robustness and generalization in ICRL**.
- Reviewers also praise the **empirical rigor and breadth** of our experiments, noting that results are “thoroughly conducted” (C1rM) and **compelling across domains** (Fyqe, cbu6).
- Notably, reviewers acknowledge the **practicality and clarity of our approach** (C1rM, Fyqe, cbu6), appreciating its relevance to the robust RL and ICRL communities.

Meanwhile, Reviewer **g2Pv** raised several concerns with a stated **low confidence** score, and we appreciate the opportunity to clarify these points. **Many of the issues appear to stem from differing interpretations of the ICRL setting and its relationship to Robust RL**, rather than from fundamental flaws in our approach. We hope that the additional explanations provided below help resolve these concerns and reflect the broader alignment among the reviews regarding the significance of our work.

## Efforts to Clarify the Concerns of Reviewer g2Pv
We appreciate Reviewer **g2Pv**’s detailed feedback.  To address these points clearly and constructively, we provide the following clarifications and supporting evidence:

**Readability, Structure, and Presentation**
- Significantly reorganized Sections 4 to streamline the narrative and **remove redundancy**.
- Added a **high-level contribution summary in the introduction** and a connecting **overview paragraph** at the start of Section 4.
- Moved all key **algorithm pseudocode** into the main text and restructured algorithm descriptions for direct readability.

**Clarification of Modeling Assumptions and ICRL/Robust RL Setups**
- Explicitly clarified that **our ICRL setting follows the standard protocol from prior work**.
- Clearly defined the adversary space and parameter set, following classical robust RL literature.
- Explained how adversaries in our framework  **naturally capture parameter shift robustness** (e.g., mass, friction), supported by *Figures 12–17*.
- Clarified the definition of robustness and that max–min training in Eq. 3 follows **established robust RL theory** and operationally yields robust policies.

**Completeness of Empirical Evaluation**
- Added extra baselines **M2TD3** and **CVaR-PPO**, expanding the baseline suite considerably.
- Highlighted and expanded **multi-domain evidence** across Dark Room, Meta-World, and MuJoCo.
- Reorganized composite figures for **improved readability**.

We believe these modifications and enhancements are sufficient in addressing Reviewer g2Pv’s concerns and clarifying any potential misunderstandings.

## Substantial Improvements Made During Rebuttal

Thankful to all the **constructive feedback** from the review committee, **we further enhanced the quality of our work** during the discussion period. Specifically, we provide

**Expanded Experimental Coverage and Stronger Baseline Comparisons**
- **Added M2TD3 across all MuJoCo tasks**, responding to Reviewer cbu6’s and g3Pv’s concern to enhance our empirical results with important robust baseline
- **Added CVaR-PPO to locomotion tasks** (Quadruped/Cheetah), further strengthening and broadening the baseline comparison in Figure 1(b).
- **Added IC2RL label-quality ablation** (Figure 8) with early-stopped, showing that IC2RL is highly sensitive to the quality of its labels. **This directly justifies our motivation for ICAG/ICAA**.

**Improved Clarity, Organization, and Self-Containment**:
- Included key **pseudocode** into the main text, enhancing algorithm clarity.
- **Relocated essential experimental details into the main body** (including adversary design, adversary strength calibration, K-selection principles, and implementation efficiency), significantly increasing the paper’s accessibility.
- **Streamlined theoretical exposition**, trimming and shifting complex proofs and analysis into the appendix to improve clarity.

**Detailed Analysis of Computational Cost and Discussion of Practical Efficiency**
- Provided a complete **wall-clock runtime table** comparing baselines with pre-training dataset generation
- Clarified that **ICAG’s dataset generation is readily parallel**, drastically reducing wall-clock time and **ICAA is extremely cost-effective**, since performance saturates within 5 iterations and rollout numbers are **minimal**.

---

Lastly, we sincerely thank the area chairs for their careful coordination and oversight of the reviewing process, especially given the unusual challenges on OpenReview this year. We appreciate the time and effort invested in evaluating our work and value the opportunity to contribute to the ICLR community.

Best,

Authors

---

### Meta-Review · Area_Chair_5D3F · 2026-01-10

**Summary:**

In this work, the authors introduce a new perspective of robust RL by connecting robustness with generalization in in-context RL (ICRL). Inspired by the DPT-like models, the paper proposes a training approach that augments the pretraining dataset with action labels obtained from robust policies, trained on diverse environment disturbances, such that the transformer can learn to generalize across tasks and perturbations. The authors then propose a series of relaxation, namely in-context adversarial generalization (ICAG) and in-context adversarial adaptation (ICAA), to avoid direct worst-case optimization and improve computational efficiency.

Main strengths:
- The connection between generalization and robustness is an interesting new finding, as noted by all the reviewers.
- The experimental results demonstrate the strengths of ICAG/ICAA in terms of nominal and perturbed performance over the ICRL and robust RL baselines across navigation (Dark Room), manipulation (Meta-World), and continuous control (MuJoCo).

In the initial reviews, the reviewers also raised the following main concerns:

(1) Definition and justification of robustness (Reviewers: g2Pv, Fyqe):

Multiple reviewers noted that the paper’s notion of robustness appears unclear, i.e., underspecified adversary capabilities, unclear assumptions, and missing connections to standard robustness concepts like parameter uncertainty (e.g., gravity or friction changes). Reviewers also found the claim that the learned policies are “robust” or “optimal” under adversarial perturbations concerning without clear theoretical or empirical support.

(2) Baselines, robustness evaluation, ablation study, and computational cost (Reviewers: g2Pv, C1rM, Fyqe, cbu6):

Reviewers highlighted the missing or inconsistent baselines (e.g., CVaR-PPO, M2TD3 in MuJoCo) and limited environments compared to the broad claims. Robustness is not clearly demonstrated, e.g., performance stability across varying environment parameters is not systematically shown. There is also a lack of ablations analyzing sensitivity to label quality and adversary strength. Reviewers also noted that the additional computational cost is not sufficiently quantified or justified, e.g., training many policies across tasks and adversaries and extensive rollout generation.

(3) Technical novelty (Reviewer Fyqe):

While reviewers acknowledged the interesting link between robustness and generalization, they also found the technical novelty somewhat incremental given that many components are standard.

(4) Theoretical guarantee and the gap between theory and practice (Reviewers: g2Pv, Fyqe):

Reviewers noted that formal robustness guarantees are missing and were concerned that the theory in the appendix depends on strong assumptions and does not adequately reflect the practical algorithms used.


Additionally, multiple reviewers found the paper hard to follow due to the verbose narrative and insufficiently explicit presentation of the core algorithm. The lack of a concise pseudocode and the reliance on appendices for key details make the main text not self-contained.

**Reviewer Concerns:**

After the rebuttal, many of the concerns (i.e., (1) and (2)) are either fully addressed or largely alleviated, but the concerns (3) and (4) still remain outstanding. Specifically,

- The first concern has been largely alleviated through a major revision that streamlined the presentation at multiple places, with pseudocode added for better exposition of the methods.

- Regarding (2), the additional experiments provided by the rebuttal addressed the concerns about the missing baselines and computational cost of ICAG, which can be largely handled through parallelization despite the need for a large number of independent SAC runs. Regarding the robustness evaluation, Figures 12-17 in the appendix were highlighted to demonstrate the empirical robustness of ICAG/ICAA to environment parameters.

- However, regarding (3), after a careful read of the paper and the rebuttal, I still share the same concern with the reviewer. At its core, ICAG appears to be essentially DPT but reframed for the robust RL setting. To be more specific, based on Equation (5), one can view the nominal MDP $\tau$ and the adversary $\phi$ jointly as the environment. By collecting in-context datasets and the optimal action labels for different $(\tau,\phi)$ pairs, the training of ICAG is fundamentally the same as that in DPT.

- As for (4), on one hand, I have reservations about the theoretical contribution. In particular, it is already known in the ICRL literature that DPT essentially does in-context posterior sampling (Lee et al., 2024). The results, assumptions, and proofs in Appendices F and G of the paper appear to be directly adapted from those in (Lee et al., 2024). Unfortunately, this is not clearly stated in the paper. On the other hand, the concern about the theory-practice gap still remains as it seems unclear if ICAA has any robustness guarantees given that the action label quality can be quite sub-optimal.

Additionally, I have another concern about the fairness of the comparison between ICAG/ICAA and the robust RL baselines. ICAG/ICAA by design require both a massive amount of pre-training data and few-shot rollouts at test time, while the robust RL baselines considered here like DR, RARL, and QARL all operate in a zero-shot manner at test time and are usually trained with much fewer data samples. With that said, such a comparison does not seem fair in terms of sample use.

To make an apples-to-apples comparison, the robust baselines shall also be allowed to use few-shot samples (e.g., for fine-tuning). Another way of comparison is to consider the robust RL methods that are able to adapt to the transition dynamics at test time, such as CaDM [1] and ReDM [2].

[1] Lee et al., “Context-aware dynamics model for generalization in model-based reinforcement learning,” ICML 2020.

[2] Jia et al., “Policy Rehearsing: Training Generalizable Policies for Reinforcement Learning,” ICLR 2024.

**Reviewer Scores:**

As several major concerns still remain, I figure that this submission is likely to still be borderline with mixed scores after the rebuttal.

The connection between robust RL and generalization of ICRL is interesting, but the design of ICAG and the theoretical contribution seem incremental compared to the DPT-like methods. Moreover, while the experimental results look promising compared to the robust RL baselines, whether the comparison is fair in terms of the sample efficiency is concerning.

For the above reasons, I am currently leaning towards rejection, but I won't mind if it gets accepted as this direction can potentially serve as an alternative to solving robust RL.

---

### Decision · Program_Chairs · 2026-01-26

Reject